# Development of Desirable Behaviors in Dog-Assisted Interventions

**DOI:** 10.3390/ani12040477

**Published:** 2022-02-15

**Authors:** Félix Acebes, Juan Luis Pellitero, Clara Muñiz-Diez, Ignacio Loy

**Affiliations:** 1Research Group on Development and Comparative Cognition, Department of Psychology, University of Oviedo, 33003 Oviedo, Spain; impronta.idi@gmail.com (F.A.); munizdiezclara@outlook.com (C.M.-D.); 2IMPRONTA Formación y Servicios Integrales de Apoyo Animal, S.L., 33510 Siero, Spain; integraprogramasterapeuticos@gmail.com

**Keywords:** dogs, dog-assisted interventions (DAI), animal-assisted interventions (AAI), training, cognition

## Abstract

**Simple Summary:**

In a recent paper, Hall et al. encouraged professionals of canine training to share their observations and procedures with researchers in the field of dog learning and cognition, with the goal of coordinating knowledge and make better use of time and resources. In response to this invitation, here, we present an integrative method for the training of dogs that take part in animal-assisted interventions (AAI). This method has been developed taking into account the needs observed during nearly 30 years of interventions for the cognitive, relational, functional, and emotional improvement of the users. This method focuses on the dog, developing in the animals the necessary skills for their inclusion in dog-assisted interventions from a constructivist perspective, while guaranteeing their well-being during the training and the execution of their tasks.

**Abstract:**

Dog-assisted interventions (DAI) are those that include specially trained dogs in human health services. Often, the training methods employed to train animals for DAI are transmitted between trainers, so the latest scientific research on dog learning and cognition is not always taken into account. The present work aims to evaluate the impact that the main theories on the evolution of the dog have had both in promoting different training methods and in the relevance of behavior in the evolution of the skills of actual dogs. Then, an integrative method for the training of dogs is presented. This method takes into account the research on dog learning mechanisms and cognition processes, and effectively promotes the development of desirable behaviors for DAI during the dog’s ontogeny.

## 1. Introduction: Dog-Assisted Interventions

Companion animals have been proven to have both physiological and psychological positive effects such as improvements in cardiovascular health, increases in motor activity, reductions in stress levels, and increased positive emotions [1,2,3]. Even untrained pets can positively impact on well-being [4,5]. Animal-assisted intervention (AAI) is defined as the inclusion of specially trained animals in services for the improvement of human health [6]. Despite the increase in the use of AAI in various therapeutic settings, the specific criteria that the animal has to meet in order to be part of these interventions, and thus, the specific training that the animal has to receive, are still not standardized. Instead, the skills, abilities or aptitudes, and the method to train them are left to the discretion of the handler. In these interventions, animals are understood as a resource that are more effective than material resources because animals react to and interact with the users [7]. Several theories have been proposed in order to explain why these interventions are effective. The animal-human bond, attachment. and biophilia can account for the improved effectiveness of multidisciplinary therapies that include animals [8,9]. Additionally, interspecific interaction can induce changes in the levels of oxytocin, which could be related to the effectiveness of this tool [10,11]. Furthermore, including interspecific interactions in therapy can focus the attention of the user in the animal [12], which can decrease possible adverse effects of the treatment such as associated anxiety and pain, resulting in an increased effectiveness of the treatment. Including animals in therapy can also reduce the therapist–patient gap, being a social mediator or a role model of socially acceptable behaviors, thus facilitating the interaction and increasing the effectiveness of the therapy [9,13]. Regardless of the particular hypothesis suggested to account for the benefits of AAI, all base its efficacy on the improvement in the contextual conditions due to the presence and the natural behavior of the animal.

AAI have been effectively implemented on a wide range of contexts such as hospitals, educational institutions, prisons, old people’s homes, and the everyday clinical practice of different health disciplines [1,14,15,16,17]. As the use of dogs in AAI became popular, the different roles that work dogs have had to play and the procedures to train them in those roles have been labelled under a variety of terms such as dog-assisted therapy (DAT), dog-assisted intervention (DAI), dog-assisted education, dog-assisted activities, support dogs or assistance dogs. Although in the last 20 years the research on canine learning and cognition has resulted in important progresses for the understanding of this species behavior, these progresses have had little impact on the training of work dogs. The training methods are usually transmitted between trainers and do not take into account the new scientific findings [18]. It is proposed that, in order to include a dog in DAI, it is only needed to develop in the animals those behaviors that, being part of the natural set of behaviors of the species, are desirable for the intervention [19]. Those behaviors that correlate with a positive evaluation in the accreditation of work dogs and that are mainly related to social education are usually trained by methods based on classical conditioning, operant conditioning, or, in a few instances, social learning [18].

Here, we present a method for the training of these work dogs that would complement the existing set of tools. This method, developed by Juan Luis Pellitero, has successfully trained dogs for their inclusion in DAI for social-health centers for the care of dependent elderly people and elderly people with Alzheimer’s disease and other dementias (residences for the elderly of the Department of Services and Social Rights of the Government of the Principality of Asturias), intellectually disabled persons with intermittent generalized or extensive support needs (residences for people with Intellectual Disabilities of the Department of Services and Social Rights of the Government of the Principality of Asturias, FASAD Foundation, ASPACE Association, and PLENA INCLUSION confederation of organizations), persons with acquired brain injury (CÉBRANO Association, COLISÉE PLAZA REAL brain injury rehabilitation unit) and in pre-primary, primary, secondary, and special education centers in the Principality of Asturias, Galicia, Castile, and Leon, the Autonomous Community of Madrid, Castile and La Mancha, Valencian Community, Mexicali and Calexico (Santa María del Naranco Alter Vía, Reina Adosinda, Pino de Obregón, El Santuario, Jaime Borrás, Enrique Alonso, Jaume I, Bajo Maestrazgo, Mission-School and HomeWork schools, and for the Ministry of Education of Valencian Generalitat and Balearic Islands Governments and the Department of Education of the City Council of Vigo). The method exploits the learning processes that the available research indicates dogs use during their ontogenetic development, promoting the natural behaviors of the species that are useful for the interventions from an approach coherent with the physiological and cognitive skills as well as the needs and the motivations of the animal.

## 2. The Method as an Educational Process

The main theories on the evolutionary process of the dog have shaped the idea of what it is to be a dog and promoted different types of training (for a review, see [20]). When the dog is understood as a wolf artificially selected by humans according to behavioral characteristics such as docility [21], the role of the dog and its behavior in its own evolutionary process is overlooked. The training methods that have been developed from this perspective are based in obedience, submission, and pack hierarchy. These methods were very popular at the turn of the century, but nowadays there are other perspectives on the evolutionary process of the dog such as the self-domestication hypothesis [22] and its ulterior developments [23,24,25]. The self-domestication hypothesis has important consequences in the understanding of canine behavior, so the model of the wolf pack, with its hierarchy based on domination–submission, has to coexist with the model of a heterogeneous group of dogs that share a feeding place, for example, a dump. Despite the concurrence of several hypotheses about the evolutionary process of the dog [26], the great impact that behaviorism has had in the understanding of animal behavior has led most of the training programs used nowadays to base their techniques in the contingent presentation of stimuli both independently (classical conditioning) or dependently of the behavior of the animal (operant conditioning), thus minimizing the cognitive skills of the animal. Regardless of the use of punishments or rewards, these methods always aim to control behavior through discrete manipulations of the context in which the dog is a passive subject. This is an effective and very fast way to control animal behavior, given that linking environmental stimuli, or even stimuli and behaviors, is a simple task for an animal with the cognitive skills of the dog, especially when compared to solving more complex tasks that dogs are able to complete [27,28]. In addition, associative learning is, at least, one of the fundamental mechanisms involved in the understanding of the world [29,30] that can modify the interspecific communicative behaviors relevant for DAI [31,32,33], although its development must be traced back to early stages of its ontogeny [34]. However, limiting the training of a dog to associative learning techniques leads to unwanted effects for the DAI, widely demonstrated in the study of animal learning. These effects include extinction, that is, the abolition of the established behaviors when the contingency (stimuli, behaviors and consequences) is not explicitly controlled, context specificity when the learning needs to be generalized to other situations, reinforcer devaluation, or focusing the attention of those who take part in the learning process (human and dog) toward stimuli, thus limiting the communicative skills and their interaction even more.

On the other hand, in the last years, similarly to other disciplines related to learning and behavior, animal training has been influenced by the cognitive paradigm in psychology, highlighting the active role of the subject in its relationship with the world and proposing explanations based on cognition to account for the behavioral results that cannot be explained from behaviorist approaches. This progress has translated to training in different strategies such as the cognitive [35,36], which suggests that associative processes need to be complemented with basic cognitive and decision-making processes in order to achieve functional guide dogs; the cognitive-emotional, which stresses the need to maintain an adequate emotional estate, given its importance in the interpretation of the presented information [37,38]; and those based on social learning such as the Do as I Do [39], which has been proven to be more effective than shaping for the training of object related actions or corporal movements [40]. All these strategies (classical conditioning, operant conditioning, emotional cognitive approach, ethological approach, social learning) are variations of an account based on unidirectional instructions using different procedures to develop a series of skills and knowledge in the dog. Even when the inclusion of cognitive elements in training is a step forward that implies a greater influence of the subject in its own learning process, it could be strengthened by integrating all of those elements in a global approach coherent with the ontogenetic development of the animal.

The debate on the best way to teach dogs parallels the debate that occurred in the theory of education regarding the best way to teach children. Until the 1980s, the most accepted model of instruction in early childhood education was based on the assumption that knowledge can be transferred intact from teacher to student. From this, educators focused on transferring knowledge to their students and educational researchers tried to find better ways to do it. However, in subsequent decades, the need to recover paidocentric educational models has arisen. In these models, the teaching and learning process focused on the student, and not the teacher. Thus, the child has to be observed in a rigorous and individualized way to make specific pedagogical decisions that will have to be progressively adapted to the evolution, expectations, and motivations of the student [41]. Similarly to the change in child education, the method presented here shifts the focus of the training process to the dog’s behavior, promoting the necessary competencies for DAI from the nature of the dog in ontogenetic development with the ability to learn through all of these processes. The use of procedures and theories from one field for the development of another has been a constant in the development of animal psychology. For example, tests from human developmental psychology were applied to primates [42] and from primates to dogs [43,44,45]. It is not surprising that our proposal parallels the historical change in education from an approach focused on the transmission of content to one focused on the child. Thus, the use of training techniques that promote the ontogenetic development of the communicative, behavioral, and social skills of the animal through within- and between-species controlled interactions have been proposed in order to develop desirable behaviors for DAI.

Since the dog is a social animal and, thus has the physical and cognitive abilities to perform coordinated behaviors in its social context, the method presented here proposes that, for its inclusion in DAI, dogs should perform the trained behaviors driven by social motivations such as affection, social facilitation, or the fulfilment of social goals and not by individual motivations such as obtaining trophic reinforcers or avoiding (escaping) aversive stimuli (prong collars or shocks). Social reinforcement is enough to increase the frequency of desired behaviors [46,47,48,49,50]. It has been found that petting leads to an increase in serum levels of social bonding and pleasure related hormones such as b-endorphin, oxytocin, prolactin, b-phenethylamine, and dopamine [51,52], which can be related to the effectiveness of social reinforcement. Even when it might not be as effective in promoting the appearance of a particular behavior on command [53,54,55], social reinforcement improves teamwork between the dog and the user or handler, and avoids side effects associated with other training methods that hamper the development of the skills needed for a successful performance in DAI [56]. Moreover, social reinforcement leads to a constant improvement in communication, which in turn benefits the development of subsequent training programs, also promoting the well-being of both dogs and humans [31]. Thus, following Friedman’s hierarchy of behavior-change procedures [57], punishment should be removed from all learning processes, not only for its inefficacy, but also for its harmful effects on the future behavior of the learner [58,59]. The International Association of Animal Behavior Consultants (IAABC) states that, taking into account the key principle LIMA (least intrusive, minimally aversive), positive reinforcement must be the primary training method as it promotes better animal welfare and less future disruptive behaviors [60].

Bray et al. [19] listed a number of behavioral features that correlate with a positive evaluation for work dog certification in general such as trust, absence of fear, low body and tactile sensitivity, and trainability (predisposition of the dog toward following orders, learning of new tasks, play search, pay attention to relevant stimuli, ignore distracting stimuli, and good response to correction). For assistance dogs in particular, Bray et al. [19] highlighted absence of aggression, chase, high levels of energy, excitability, hyperactivity, leash pulling, reactivity, and barking or vocalizing. The method presented here develops in the dogs these technical resources and social abilities, selecting the dogs that display behaviors that predispose them to the acquisition of skills, abilities, and aptitudes that are significant for the professional future of the dog, and controlling, during their training, the environmental factors that allow for their development. For this reason, the trainer needs to have better knowledge on dog cognition in general, and of their perception and reaction to human emotions in particular, which would result in an improved well-being not only of the dogs, but also of the humans that live with them [61].

## 3. Dog Selection

The hypothesis of self-domestication of the dog not only has important consequences for the ethological model of reference for the understanding of the behavior of the dog as above-mentioned, but also to the relevance of behavior in evolution. Thus, from this account, wolves that began to live close to humans imposed new conditions of selection that favored smaller size, more rounded features, smaller canines, tolerance to other scavenger wolves, or the ability to interpret human behavior [22]. This behavioral change that drives evolution could be an example of organic selection [62,63,64]. It could also be argued that similar characteristics could appear, even faster, from wolves taken from the wild and artificially selected by humans [21,26]. Either way, dogs are very skilled in both understanding and producing communicative signals [65,66,67,68], consistently communicating with the human for attention and help [69,70,71,72]. From birth, dogs are exposed to communicative and cooperative human sounds, gestures, expressions, and interactions that create numerous opportunities for them to learn from humans [73,74]. Bidirectional communicative acts are commonly reinforced throughout the life of the dog, thus increasing their occurrence [75] and adaptability to the environmental contingencies [76,77]. These communicative skills enable dogs to discriminate between facial expressions [78,79,80,81] and diverse sounds and verbal messages [82,83,84], integrate the information simultaneously obtained from different sensorial pathways [61,85], and use human emotional expressions to guide their behavior [86,87,88]. All of these lead to dogs looking at the human face to search for help in conflicting situations, for example, when the reinforcement is out of reach [89].

Since artificial selection has rapidly accelerated the phylogenetic transformation of the species and of the morphological or behavioral features that the different breeds and crossbreeds develop during their ontogeny, the breed influences the sensory skills (e.g., [90]) and, thus, the behavior of dogs. Regarding behavior, each animal would display a tendency to use, most often, a particular sensory modality in its relation with the world. If the DAI requires an animal that successfully understands and predicts the human behavior (user, trainer, therapist, agent of socialization…), then the best strategy is to select those subjects that have the same preferred sensory modality, in order to ease communication. Given that humans are visual animals [91,92], it would be very important to select those dogs that would preferably use vision to perceive their surroundings, infer the future, and make decisions congruent with their goals. This reduces the communicative mistakes due to traces, sounds, or rapid movements in the peripheral vision that can appear when using breeds artificially selected and employed for purposes other than support or intervention, and maximizes the prospects of success in the bidirectional communicative act that takes place during DAI. Research has shown that some work dogs show a higher sensitivity to human visual and vocal stimuli [93] and that DAI dog owners are more successful than domestic dog owners in understanding and inferring the behavior of their dogs [94]. It is also worth noting that, even when humans and dogs are evolutionary distant species, the use of mutual gaze as a communicative tool has great relevance, related to an increase in the oxytocin levels in the humans that stare at their dogs as well as in the stared dogs, thus enabling mutual affiliation [95].

Some dogs from breeds included in the Group 8 of the Federation Cynologique Internationale (FCI) such as the golden retriever or Labrador retriever, understand humans particularly well and also naturally exhibit specific behaviors relevant for DAI due to their specific features. The preferred selection of these breeds does not intend to exclude any dog belonging to any of the other breeds recognized by the FCI nor mixed-breed neither shelter dogs, but aims to choose animals that have a higher probability of displaying specific relevant behaviors for the role they would play in the DAI as well as maximizing the control over the ontogenetic experiences of the dog, given their importance in the development of canine social cognition [96,97] and interspecific communication [31]. Shelter dogs can develop, during their ontogeny or after appropriate training, the needed skills to be a DAI dog, and it has even been proven that the limited contact with humans in the daily activity of shelter dogs might result in an increased motivation to approach or interact with humans, spending more time near unknown humans in sociability tests when compared with family dogs [98]. However, due to past experiences, they tend to show behaviors incompatible with DAI such as fear-appeasement responses (tail and ears down), which is known to have an inverse correlation with other desired behaviors such as persistently gazing at the human [99]. Furthermore, shelter dogs usually have a worst performance than family dogs in responding to stimuli that predict the attentional state of humans [100] as well as in a pointing test with complex signals [96]. Nevertheless, these deficits could be compensated for with additional training. Even when animals from the Group 8 of the FCI (Labradors) are compared, shelter dogs show differences with family dogs in their problem-solving strategies and communicative skills, shelter dogs taking less into account humans and their gaze when trying to solve an insoluble task [101]. Research on visual communication shows that DAI dogs, whether due to training or to their interaction with more humans, gaze at their owners more often in visual attention tests [102], spend more time gazing to obtain inaccessible food [103,104], and alternate their gaze more between an insoluble task and their owner [94], thus being able to better communicate with humans than other dogs. Topál et al. [39] reported that highly trained dogs can successfully point to the location of a box with bait and the key needed to open it, while other dogs can successfully signal the location of a desired toy but not the location of the stick needed to retrieve the toy ([105]; for a review on the effects of training on the improvement in the ability to understand the social behavior of the human, see [30,59,98,106,107]).

Nonetheless, the behavioral traits critical for the success of the working dog are directly influenced by the genes the subject carries and its life experiences (maternal care, experiences in the juvenile period, and even the breeder’s experience). Thus, choosing golden retriever or Labrador retriever pups from small-scale breeders committed to the importance not only of the morphology and the production, but of behavior in the breeding process [108], might be an option that would entail the lowest economic and temporal investment as well as the most socially responsible, avoiding the indiscriminate breeding and rejection of many pups [19]. Lack of knowledge on the origin of the dog and the possible physical and psychological problems it might have is a risk that might impair the bond between the dog and the users or handlers by unforeseen problems (avoidance, fear, hip or elbow dysplasia, skin diseases, generalized tonic-clonic crisis...).

Furthermore, the method proposes to choose, from the available pups, those more predisposed to follow an unknown human, that show cordial interaction with their littermates, take part in the social cohesiveness of the group, avoid open conflicts for the available resources and show interest to novel sounds emitted by humans, approaching to identify them, and not showing fear or avoidance. The display of these behaviors between the fourth and eighth week of life of the pup is not a complete guarantee of the success of the training of the DAI dog as it does not test specific behavioral or cognitive skills, but demonstrates that behaviors related with the essential abilities for working dogs [19] can be identified in the selected pups, at the beginning and the end of the critical socialization period [109,110,111], with observation criteria that can be operationalized in order to design experiments that might lead to new developments in the field of selection and training of DAI dogs.

## 4. Skill Training

Within this dog focused method, basic training starts around the tenth week. This training promotes in the dog the skills and technical resources needed to live in society and establishes the rules, limits, and behavioral patterns required to establish trust, understanding, and respect between dogs and humans. For a DAI dog to have a successful, efficient, and functional basic training, this should be directed toward subsequent specific training. In the course of basic training, the necessary skills for further training are developed, thus employing guidelines different to those employed in the training of championship or pet dogs, and congruent with the final goal of achieving a dog that can support the promotion of the personal autonomy of humans with functional diversity. In this phase of imprinting and socialization, situations that promote the appearance of behaviors desired for a future DAI dog are enabled or favored, aiming to improve, with our behavior, the occurrence probability of these behaviors along the training and in the adult life of the dog.

The first basic skills that the DAI dog has to acquire are following the human and come when called. The first weeks of the ontogenetic development are crucial in translating the natural behavior of following its social group into accompanying the user, trainer, or human the dog is with at any moment. To train this, it is enough to start walking with the dog (with varying speed, according to a calmed pace for the pup), whether with an extensible leash (with no tension) or without a leash, if the training takes place in a totally controlled environment, without flight risk. Over the course of this training, the pup often initiates normal exploratory behaviors, being that its attention is attracted by almost every visual, olfactory, and sound stimulus that it meets. However, a well selected pup would not cease to pay attention to the trainer for a long period of time and, if the distance with the trainer is too far, the pup would reduce it with a short run. When the pup reaches the trainer, they should direct their attention to the dog, socially rewarding the behavior by petting the dog (from the neck to the tail), doing movements and verbalizations that indicate joy, and even making (if possible) the dog pass between their legs without stopping the walk. During these trials, the dog might be attracted by some object, animal, or plant and decide to focus on identifying it. In these cases, the trainer can try to gain the dog’s attention with high-pitched sounds and/or movements that are interesting for the pup, so it initiates a run toward the trainer that should be rewarded, as previously stated. These trials allow the trainer to start to develop in the pup the future behavior of come when called. The name of the dog can be used to call it as long as it responds to it and all the trials end with the behavior that is being trained, otherwise, its own name ceases to have any significance for the dog. In this phase, it is crucial that every trial ends with the pup going to the trainer, and not the reverse, as the training would not develop the desired skills but others such as initiating a following game or inhibition of human contact. The use of louding, that is, capturing the dog’s attention with a particular stimulus (object, movement, sound) that is interesting for the dog in order to guide its movement can ease the task of the trainer. As the training progresses, objects, plants, and even conspecifics and other animals (including humans with varied behaviors) that attract more attention are included in the training trials, as long as it is reckoned that all trials would finish with the pup paying attention to the trainer and making the decision to follow them in their walk. This training and real-life situations with comparable conditions are repeated throughout the dog’s life, so it is not necessary to establish a particular number of replications, but only work when trials in which the goal is not met are observed. The come when called direction is completed with the opportunity for the user or handler to put the dog on the leash. On the other hand, the following direction is completed with the development of training toward leash-guided walking, looking to establish behavioral synchronization with the user or handler, and discrimination of obstacles and other stimuli. Training trials have to be spaced in time to avoid establishing a routine and allow for the ontogenetic development of the animal, which enables variations in the behavior [112] and extends the conditions in which they take place [113].

Allowing the dogs to play with conspecifics in-between training trials, both in the presence and absence of the trainer (for the effect of the trainer in play, see [114]) but always under controlled conditions, favors the socialization of the dog with other members of their species. These conspecifics can be socially skilled dogs, pups of the same age and/or, even better, dogs that have already been trained in the skills that are being trained (modulating adults). Furthermore, these free play intervals help the dog learn to discriminate between leisure and work times as well as allowing the trainer to observe the natural behavior of the pup in a group of peers, so the behaviors (desired and undesired) that the dog starts to display can be analyzed and decisions on the next steps of training can be made. Free play, which can be used as a reward for the pup, and the other trainings that involve physical activity are the best way to tire the pup, with the subsequent emotional state of relaxation. Both are useful to create the optimal situation for the training of many DAI related skills such as sit and lie down.

In its daily life, a DAI dog has to be still for a long time, maintaining a nearly null level of activity (stays), independently of the context. Thus, for the dogs, a stay cannot be an obligation or a behavior learned in response to an order, given that they are long, intermittent, with varied etiology and characteristics, and, above all, beyond its control (independent of its emotional state). Therefore, it is necessary that the dogs develop the skills needed to appropriately manage these situations, being able to obtain the maximum possible self-benefit, so these experiences are significant for them and, consequently, are implemented in the animal’s lifestyle. To train stays, it is enough to stop next to the dog and stay put after any of the aforementioned trainings or a play session. In this new situation of absence of movement next to the trainer and without distractor stimuli that would make the dog start walking, the physical fatigue makes the pup decide to sit, at first, and/or eventually lie down, if the situation is prolonged on time, so both behaviors can be contingently reinforced. It could be argued, as in many other behaviors, that operant conditioning is increasing the occurrence probability of the behavior (the dog sits and receives a pet) and this might be partly true, but from our approach, it is the behavior of an intelligent being that, from a wide variety of possible behaviors in a particular psychophysiological situation, decides to assume a pose that, in addition to offer a chance for rest and relaxation after physical effort, is socially rewarded and appreciated by the group, thus being highly relevant and significant for the pup. This is why not many trials are needed for the pup to consider sit/lie down as a possible behavior and (in the absence of contrary experiences) the most relevant to display in all of the situations in which the dog transitions from movement to its absence, not needing any more discriminative stimuli (orders or commands). A dog that has been trained in this way would sit/lie down during a stop in a pedestrian crossing or by a bench, as it knows it can and its groups likes it to do so. This behavior would also be displayed when arriving at a classroom, consulting room, or other rooms (after entering a building and go upstairs), if (after numerous previous experiences) the dog foresees that it would need to save energy in order to face the working session ahead. It might happen that, at first, it would not decide to sit/lie down because it is not tired or prefers to stand up to better or farther smell or see, but if the situation is maintained and the relevance of other possible behaviors disappears, lying ends up being the only significant alternative, being chosen and reinforced due to its physiological relevance (in a fixed fashion) and by social reinforcement (intermittently), with gazes, “well-done”, “good dog”, attention… As trials progress, the pup grows and its physical resistance increases, so deciding that sitting is the best option occurs in a lower number of occasions. The latency to complete the training routine increases (especially on the trials in the beginning of the working session), so the trainer can use the strategy of stepping on the leash in stops (discriminative stimulus) to indicate to the dog that sitting/lying down is the more appropriate behavior because the stop is going to be long or just because it is what is required from the dog at that moment. The course of training eases the understanding of the message and its natural relevance preserves the behavior in the newly generated context (stop with stepped-on leash). The manual relocation of the pup (from sitting/lying) in order to optimize the posture and adapt it to the situation (e.g., stays in public transport), far from contraindicated, is yet another opportunity in training to habituate dogs to manipulation, a very important skill for DAI dogs. However, the goal of keeping the pup in the sitting/lying position is the priority and, even if the dog stands up or shakes when it is being manipulated, the trial must end in the established position, being that the only possible disadvantage is having to prolong the trial until the goal is achieved. Manual relocation such as other behavior optimization techniques that can be used during training (e.g., shaping) is almost unnecessary if modulating adults, whose behavior can be imitated through social learning [115,116,117,118] are available, minimizing the number and duration of trials.

Most of the stays that DAI dogs perform need to take place in height to favor a comfortable interaction with the users. For this reason, from the first trials of training, pups must be invited to climb to, with help, or gently lifting them if necessary, and use a pedestal as the preferred resting place. This, as happens with the stops, makes the dog understand, after a generalization process through successive approaches, that flat surfaces in height (chairs, benches, geriatric chairs, electric stretchers…) are an ideal place to rest. The development of the stay in height behavior in the pups promotes their proprioception, at the same time, it provides the trainer with a new tool for stimuli control by providing a new, rest and relaxation related context that naturally restricts the activity and movements of the pup. This layout favors the trials of social learning by observation, of inhibition to motivational elements ([119,120,121,122,123]; for a review see [124]), and of resting manipulation to inhibit contact reactivity, also favoring cleaning, brushing, and sanitary intervention habits.

Dogs form deep attachment bonds with their owners [125] and show differential behaviors to humans with different degrees of familiarity [126]. However, the method presented here aims to establish this bond with the human figure and not with a particular human (the trainer), thus facilitating the establishment of future bonding units with the handlers and other humans in the DAI. Dogs with secure attachment to humans divide their attention between the user and the handler, whereas dogs with an insecure attachment focus only on the handler ([127]; for further development see [18]). For this purpose, during the pups’ training, different socialization agents need to be used. This role would be played by different people (trainers or not) that, knowing the training method that the dog is completing, commit to taking care of the pup in its daily life for one or two weeks, following the directions given by the trainer. The use of multiple agents of socialization enables the generalization of the skills developed during the training to other contexts, independently of the human that is with the pup, diversifying the experiences that the pup lives, and generating new opportunities for the interaction of the pup with other humans, thus promoting a secure attachment with the human.

Finally, in this stage, it is necessary to increase the communicative intention of the pup with the human by promoting play situations in which the pup is headed to the face of the trainer or stimulates different parts of the body with its tongue (even orofacial stimulation in the case of medical alert dogs), so once these natural behaviors appear, they can be rewarded. These behaviors are highly desirable in DAI dogs as, for instance, the stimulations increased the effectiveness of the intervention in cases of dystonia and muscular rigidity [128]. Additionally, avoiding eye contact with humans leads to unwanted effects in therapy [18].

The method aims to develop, from the wide range of natural behavioral skills and abilities the dog has, those that are useful for the interventions. This is achieved by promoting the occurrence of those behaviors that are useful for the dog and the users, while removing the functionality of unwanted behaviors, despite the communication problems between dogs and users. Saying “NO” to a dog, in an attempt to stop it from doing something that the dog has already decided to do, at best, has no meaning for the dog, so it will maintain that behavior and following ones. At worst, it could avoid the appearance of other behaviors that the dog has already decided to perform. Given that it is impossible to know what the dog is thinking while performing a particular behavior, discrete ideas cannot be corrected, and it is better to ignore that behavior (not give it a functional significance for the dog’s life) and let the rest of its thoughts translate into behaviors that would be promoted if appropriate.

In order to decrease the probability of occurrence of a behavior, the method relies on the control of the context and the promotion of incompatible behaviors while ignoring the appearance of unwanted behaviors in order to make them irrelevant for the dog. For example, if the pup bites a table, this has to be ignored and the dog is given a toy it can have in its mouth when the table is present and/or remove the table until the pup is sufficiently developed to not think or decide to bite it. In turn, the increase in the probability of the occurrence of desired behaviors would be favored primarily by the contingent presentation of tactile stimulation and pleasing verbalizations and gestures [49]. The use of both types of stimuli allows us to reward at a larger distance and with a more precise contingency, in addition to reducing adverse effects such as satiation or reinforcer devaluation by using various reinforcing stimuli [129].

Once the desired skills are acquired in basic training, which usually extends to the seventh month of life, the specific DAI training is initiated by habituating the pup to work contexts, generalizing to them the previously acquired skills and developing from these the specific skills for the dog’s work. At this stage, each DAI dog displays its own strengths and weaknesses and the trainer has to take them into account in order to adapt the training to the needs of the target groups [19]. The previously acquired skills are developed so an appropriate bonding unit with the handler or the users would be established. DAI dogs, unlike the dogs that are familiar with a conventional use of the leash, have to get used to uses of the leash adapted to the needs of their work, pass through doors with different strategies depending on the user they go with (wheelchairs or guided ambulation), climb on the furniture in work centers against the usual educational rules, stand on their hindquarters in different situations during sessions, etc. In such a varied environment, discrete training can lose its effectiveness [112] while a flexible basic training, focused on the cognitive development of the dog, guided by the needs that the dog has to cover in its adulthood and is sustained by rewards that are widely and frequently present in the final working environments, allows the constant development of new skills during the daily life of the dog, this being our final goal in the training of DAI dogs.

## 5. Conclusions

DAI are effective due to the presence and the natural behavior of the dog, which promotes the development of the optimal conditions for human therapy, maximizing its effectivity. Thus, training non-natural skills is not needed for DAI, but only those natural behaviors of the species that are desirable for the intervention such as trust, absence of fear, low body and tactile sensitivity, trainability, and extinction of unwanted behaviors such as aggression, chase, excitability, hyperactivity, leash pulling, and barking.

The dog focused method that has been presented here is an effective and minimally intrusive tool to develop significant behaviors from the range of natural skills and abilities of the dog by means of within- and between-species interactions in controlled contexts. The method relies on all the learning mechanisms and cognitive processes that have been proven that dogs use in their learning, and favors the establishment of desirable behaviors along the entire ontogenetic development of the animal.

Furthermore, using solely social reinforcers such as affection, social facilitation, or the fulfilment of social goals allows for the situations inherent to the intervention context to reinforce the behaviors that the dog has developed during training. In turn, this preserves the affective bond of the dog with the human figure, establishes bonding units with their handlers, and improves the interspecific relationships of DAI dogs. Thus, the method promotes the continuous development of communicative, behavioral, and social skills, in turn minimizing unwanted effects such as extinction, salivation, or handler-directed attention, given that it does not rely on discrete trainings based on individual motivations such as obtaining trophic reinforcement or avoiding aversive stimuli.

All in all, this educational approach is presented as an integrative training method that successfully trains dogs for DAI, while ensuring that the dog is behaving as an individual of its species, feeling rewarded by it, making decisions instead of following orders, developing cognitively while doing so, and retaining a good quality of life.

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
