# Peer review of "Development of Desirable Behaviors in Dog-Assisted Interventions"

_animals, 2022, doi:10.3390/ani12040477_

Round 1

Reviewer 1 Report

Dear Authors,

This is an interesting approach to training DAI dogs. The literature review is thorough and the importance of ensuring optimal dog welfare for working dogs is important for both the dogs as well as their human partners. A few areas that need to be addressed:

I would suggest a review and editing of language for clarification purposes. For example line 55 "deprivation of liberty facilities" I assume you are referring to prisons? The language could be clearer throughout the manuscript.

Lines 269-272 Do you mean domestic dogs vs other canid species or family dogs vs other domestic dogs? This is unclear to me.

Line 279 "the experiences that lives" ? I am unsure what is meant by this.

In several places you state that there is scientific consensus on the self-domestication hypothesis (e.g. Line 105). However, this is not true.  There is still debate regarding domestication between two theories: commensalism  (self-domestication) and cross-species adoption (pet-keeping). For a recent discussion see Commensalism or Cross-Species Adoption? A Critical Review of Theories of Wolf Domestication by James Serpell in Frontiers in Veterinary Science April 2021. While I don't believe this is reason to reject this paper I do believe you need to edit to reflect the lack of consensus on the domestication process. 

Author Response

We would like to thank you for your comments and revisions. We have edited the language as suggested:

  • In line 56, where it said “deprivation of liberty facilities”, now it reads “prisons”.
  • In lines 272-274, there was a mistake where we referred to domestic dogs, and now it reads DAI dogs. In addition, the reference 102 has been corrected.
  • In line 283, where it said “the experiences that lives”, now it reads “its life experiences”.

In addition, the whole paper has been revised in order to better reflect the lack of consensus on the evolutionary process of the dog, and included the reference you mentioned as reference 26, in line 106.

Reviewer 2 Report

Although the paper presents an adequate summary of current thinking on dog domestication and training the suggestions for training approaches for DAI dogs are neither innovative nor supported by any empirical or quantitative comparisons of the effectiveness of different methods, nor does the paper offer any well-defined measures of success of DAI training or the negative effects of other approaches.

The central assertion that "the method presented here develops in dogs these technical resources and social abilities". That is the hypothesis, now prove it.

The lengthy theoretical discussion should be condensed into concise practical suggestions for selecting, evaluating and training DAI dogs that can serve as the basis for gathering real-world data to document the value of this approach.

Author Response

We would like to thank you for your comments. Regarding the lack of innovation on the training approach, to the best of our knowledge, constructivism and paidocentrism have never been related to the training of assistance dogs. Therefore, the use of training techniques which promote the ontogenetic development of communicative, behavioral and social skills in the dog is innovative, or at least has never been presented as an explanatory approach, as the manuscript does. This educational approach is presented as an alternative, and it has been successfully used for the training of DAI dogs for more than 30 years, achieving dogs that have integrated in a wide range of DAI, as depicted in lines 72-88. However, measuring the effectiveness of the different methods is challenging, given that there are not specific criteria that the animal has to meet and reason for the effectiveness of DAI is still debated. It is worth noting that other approaches present well documented negative effects such as extinction, context specificity, reinforcer devaluation or excessive attention towards reinforcers. It should be noted that the present manuscript does not aim to defend a hypothesis, but to analyze a workspace, review the evidence on the cognitive abilities of dogs and present a method that has never been published, but been successful in the training of DAI dogs, hoping the scientific community would find it interesting to evaluate. 

Reviewer 3 Report

The submitted commentary provides an in depth review of an important topic, showing a novel way of interpreting the scientific literature, it contains reasoned arguments, is concise, well argued, and erudite.

Minor revision is required:

The aim of the paper should be clearly stated (it is missing in the current manuscript).

Author Response

Thank you very much for your comment. We have re-written the simple summary and abstract sections, trying to state more clearly in the simple summary the aim of the paper. 

Round 2

Reviewer 2 Report

The revisions have improved the grammatical and other errors but do not change the fact that this is not a data based or experimental paper. It would likely be of interest to other less experimental-based behavior journals, but I still feel the approach is not appropriate for Animals.